# Cytological Quantification of Nodal Mast Cells in Dogs Affected by Non-Neoplastic Condition and Mast Cell Tumor Using Different Sample Preparation Techniques: An Explorative Study

**DOI:** 10.3390/ani13162634

**Published:** 2023-08-15

**Authors:** Giulia Buzzi, Matteo Gambini, Camilla Recordati, Valeria Grieco, Damiano Stefanello, Roberta Ferrari, Clarissa Zamboni, Martina Manfredi, Chiara Giudice

**Affiliations:** 1Dipartimento di Medicina Veterinaria e Scienze Animali, Università degli Studi di Milano, 26900 Lodi, Italy; giulia_buzzi@yahoo.com (G.B.); matteo.gambini@i-vet.it (M.G.); camilla.recordati@unimi.it (C.R.); valeria.grieco@unimi.it (V.G.); damiano.stefanello@unimi.it (D.S.); roberta.ferrari@unimi.it (R.F.); clarissa.zamboni@unimi.it (C.Z.); martina.manfredi@unimi.it (M.M.); 2AniCura Istituto Veterinario di Novara, Granozzo con Monticello, 28060 Novara, Italy; 3I-Vet S.r.l. Diagnostica Veterinaria, 25020 Flero, Italy

**Keywords:** dog, mast cell tumor, cytology, lymph node, staging

## Abstract

**Simple Summary:**

Cytological evaluation of lymph nodes (LN) during staging of canine Mast Cell Tumors (MCTs) is a hot topic. While histology is currently the gold standard of LN evaluation, cytology would have the advantage of in vivo investigation that allows for the planning of a therapeutical approach. However, at present, LN cytological examination and interpretation needs to be standardized. In the present work, the significance of the nodal mast cell number in dogs with and without MCT has been investigated, comparing different counting methods and different sample preparation techniques. Our results suggest that, while counting methods and sample preparation technique do not influence MC count, the nodal MC number can discriminate between metastatic and non-metastatic LNs, but fails to distinguish between metastatic and possibly metastatic LNs.

**Abstract:**

Cytological evaluation of lymph nodes (LN) in canine cutaneous mast cell tumors (MCT) has a key role in MCT staging. However, cytological discrimination between metastatic and reactive LNs is debated and diagnostic criteria inconsistent. The aim of this study was to retrospectively quantify nodal mast cells (MCs) in non-oncological (NOD) and MCT-bearing dogs (MCTBD), using different sample preparation techniques, to evaluate the significance of the MCT number. Cytological specimens from NOD-LNs (10 fine-needle aspirates—FNAs) and MCTBD-LNs (10 FNAs, 10 scrapings, 10 touch imprints) were evaluated. MCTBD-LNs were grouped in: non-metastatic, possibly-metastatic, and metastatic based on current literature criteria. MCs were counted in 4, 8, and 20 high-power-fields, and over 500, 1000, and 2000 total cells. MCs were significantly more numerous in MCTBD-LNs than in NOD-LNs and in “metastatic” samples than in “non-metastatic”. There was no significant difference between “metastatic” and “possibly metastatic” samples. Sample preparation techniques did not influence these results. A negative correlation between MCs number and sample cellularity was observed. Results were confirmed regardless of the counting method applied. MCs counting per se cannot distinguish possibly metastatic and metastatic cytological samples. Sample preparation technique and the counting method applied seem to have no influence on cytological quantification of nodal MCs in MCTBDs.

## 1. Introduction

Mast cell tumor (MCT) is the most common cutaneous neoplasm in dogs [1,2,3,4,5]. MCT biological behavior is extremely variable [1,2,3,5,6,7,8], and metastases, when present, are most frequently seen in the lymph nodes, with possible spreading to distant sites such as the spleen, liver, and bone marrow [9,10,11,12,13].

Lymph nodes evaluation for determining metastatic status is, therefore, highly recommended [1,5,14,15,16,17,18]. Histological evaluation of the entire LN is currently considered the gold standard test to evaluate nodal metastatic status in canine MCT [15,17,19,20,21,22,23]. Nonetheless, the evaluation of LN metastatic status via cytological examination of fine-needle aspirates (FNA) before lymphadenectomy, if reliable, could be important for planning or ruling out the extirpation of regional/sentinel lymph nodes [3,5,9,12,14,15,20,21,22,24,25].

Unfortunately, cytological interpretation of LN in canine MCT is still controversial [5,21]. The literature is extremely fragmentary and reported to be highly variable, with sometimes subjective and non-specific cytological criteria. In 2009, Krick and coauthors [15] proposed a cytological interpretative system for the evaluation of LN metastatic status in canine MCT, with strictly codified cytological criteria leading to five categories that were statistically correlated with the clinical outcome/median survival time, i.e., normal LN, reactive lymphoid hyperplasia, possible metastasis, probable metastasis, and certain metastasis [15]. Krick’s interpretative system represented a key turning point in the cytological evaluation of nodal samples from MCT-bearing dogs [5]; however, with a few notable exceptions [12,23,24,26], it was mostly inconsistently applied in successive works [16,18,20,22,25,27,28,29] or underwent main modifications [21,22,30,31].

At present, no clear consensus exists regarding the quantity of MCs that can be present in normal or reactive canine LNs in dogs [2,9,16,21,30,32,33].

To the best of our knowledge, only four studies so far included the quantification of MCs in LNs from MCT-bearing dogs [9,21,23,30]. Still, it is not clear whether an increased number of well-differentiated MCs in LNs of MCT-bearing dogs should be attributed to the chemo-attractive effect of the primary mass microenvironment on reactive cells or to an effective migration of neoplastic MCs draining from a well-differentiated primary mass [14,15,16,24,27,34]. Additionally, no easily evaluable morphological features are available to distinguish well-differentiated neoplastic MCs from reactive or inflammatory ones [5,10,14,21,24,26,27].

More standardized and feasible criteria for LN cytological assessment that would correspond as much as possible to the histological classification by Weishaar et al. are, therefore, still needed [16,17,21,28].

Moreover, the growing attention toward LN mapping techniques in veterinary medicine has pushed forward the need for standard criteria in interpreting lymph nodes metastatic status [16,25,35,36,37,38,39,40,41,42,43,44,45,46].

The purpose of the present study was, therefore, to perform a preliminary investigation to verify whether the quantification of nodal MCs is a useful feature per se to evaluate the presence or absence of MCT nodal metastasis in canine cytological samples, regardless of the sampling method applied to obtain the specimen. The number of nodal MCs in MCT-bearing dogs (MCTBDs) and dogs affected by non-neoplastic conditions (NODs) was compared. Different sample preparation techniques (i.e., fine-needle aspiration, scraping, or touch imprint) were also compared to verify if the sampling method significantly impacts on the number of nodal MCs in samples from MCT-bearing dogs.

## 2. Materials and Methods

### 2.1. Case Selection from the Archives and Preliminary Classification into Clinical Categories

The database of the Diagnostic Pathology Service of the Department of Veterinary Medicine and Animal Science (DIVAS—Università degli Studi di Milano) was retrospectively searched for canine cytological nodal samples collected between 1 January 2016 and 31 December 2018. Cytological samples obtained in vivo during daily clinical activities had been submitted by the University Veterinary Teaching Hospital and by external private veterinary clinics. Additionally, cytological samples were prepared by the authors ex vivo from surgically extirpated LNs sent for histopathological evaluation to assess regional or, more rarely, systemic/multicentric lymph-adenomegaly conditions, or for neoplasm staging. Most samples prepared via scraping, smearing and/or touch imprinting were obtained from clinically non-palpable SLNs surgically extirpated from MCTBDs (Appendix A).

All cytological slides were air-dried and manually stained with May-Grünwald-Giemsa (Merck KGaA, Frankfurt, Germany).

Each cytological sample was considered as a single independent “case” even if belonging to a different LN from the same dog or belonging to the same LN sampled with different techniques. According to the cytological diagnosis reported in the database as well as to the information contained in the medical record of each dog, cases were classified as: NODs (cases from non-oncological dogs, e.g., dogs with reactive, degenerative, or infectious-inflammatory conditions) or MCTBDs (cases from MCT-bearing dogs).

Cases were excluded from the study if belonging to dogs lacking detailed clinical history, or to dogs affected concurrently by MCT and other neoplasms.

Only cases obtained by fine-needle aspiration (FNA) were included for NODS, while cases from MCTBDs were further classified based on the sample preparation technique: MCT-F (FNA), MCT-S (scraping smearing), or MCT-T (touch imprinting).

Each case was considered eligible for the study if at least 1 cytological slide was available for review, and if at least 20 adequately cellular high-power fields (HPF) were available. Each microscopic HPF evaluated with routine light microscopy (Olympus BX51TF; Olympus Corporation, Tokyo, Japan) was considered as “adequately cellular” when at least 50% of its surface was represented by well-preserved cells arranged in a single layer without overlapping.

Among eligible cases, the 10 most recent ones for the NODs category and for each subcategory of MCTBDs (i.e., MCT-F, MCT-S, and MCT-T) were arbitrarily selected, for a final caseload of 40 cases (Appendix A).

Details regarding each LN included in the study (including histopathological diagnosis according to Weishaar, when available) and signalment data of dogs were reported in Appendix A.

For each case obtained from MCTBDs, the metastatic status was cytologically evaluated applying the criteria proposed by Krick et al. [15] modified based on Mutz et al., 2017 [21]; and Fournier et al., 2020 [36], as detailed in Table 1. Specifically, cases were classified as “non-metastatic” (MCT-NM), “possibly metastatic” (MCT-PM), and “Metastatic” (MCT-M).

### 2.2. Criteria Applied to Microphotograph Acquisition and Counting

For each investigated case, microphotographs of 20 hotspots, adequately cellular HPFs (each one corresponding to an area of 434.97 × 327.51 microns) were taken with a digital camera Olympus DP70 (Olympus Corporation, Tokyo, Japan) supported by the software Olympus DP-Soft (v.5.0 for Microsoft Windows; Olympus Corporation, Tokyo, Japan). “Hot-spot” was defined as those characterized by the best balance between cellularity and cell preservation.

Each microphotograph was then randomized and blinded for category, subcategory, and signalment information. Microphotographs were then uploaded in the Fiji software 1.8.0_172 [47] and manually counted after application of a square grid. The number of MCs for each evaluated case was then determined applying 6 different counting methods: “per fields” (i.e., number of MCs in 4, 8, and 20 HPFs) and “over cells”: (i.e., number of MCs over 500, 1000, and 2000 cells counted at 400×). Naked nuclei and cells in mitosis or only partially depicted in the pictures were not counted (Appendix A). To avoid introduction of bias during counting “over cells”, each microphotograph used for this purpose was entirely counted. When cellular counting overwhelmed the threshold of 500, 1000, and 2000 cells, values were proportionally recalculated.

### 2.3. Statistical Analysis

For each counting method applied to each category and subcategory (from now on indicated as “groups”), both the median (with minimum-maximum range) and the mean (with standard deviation) absolute and percentage number of MCs were calculated to facilitate comparison of our results with those reported in previous studies.

Data distribution for each counting method applied to each group mentioned above was investigated with the Shapiro-Wilk normality test. Considering that the majority of datasets did not pass the normality test, non-parametric tests (i.e., Mann-Whitney test for pairwise comparison and Kruskall-Wallis test with Dunn’s correction for multiple comparisons) were applied to investigate the difference in the median number of MCs for each counting method among the following groups: NODs versus MCTBDs; MCT-NM versus MCT-PM versus MCT-M; MCT-F versus MCT-S versus MCT-T. Additionally, the error coefficient (obtained dividing the standard error of the mean by the mean) for each counting method applied to MCTBDs was calculated.

The difference in the mean cellularity among the different groups was also investigated. Mean cellularity of each case was defined as the mean cellularity observed in the first 5 microphotographs entirely counted during counting “over cells”. Although all the datasets reporting mean cellularity data passed the Shapiro-Wilk normality test, non-parametric tests (i.e., Mann-Whitney test for pairwise comparison and Kruskall-Wallis test with Dunn’s correction for multiple comparisons) were applied to maintain consistency with the other statistical analyses. Finally, to verify whether cellularity influenced the number of nodal MCs, the Spearman correlation coefficient was calculated.

Statistical analyses were performed with GraphPad Prism software (version 9.0.0 for Windows 64-bit, GraphPad Software, San Diego, CA, USA, www.graphpad.com). Statistical significance was set at *p*-value ≤ 0.05.

## 3. Results

### 3.1. Selection Workflow and Signalment Data for the Cases Included in the Study

Four-hundred ninety-two (492) electronical records of canine LN-cytological specimens collected between 2016 and 2018 were revised (Appendix A). Twenty-seven NODs samples, 13 MCT-F, 25 MCT-S, and 32 MCT-T samples were considered eligible for the study, for a total of 97 specimens, among which the 10 most recent cases for the NODs category and for each MCTBDs subcategory were selected, for a final caseload of 40 cases (Appendix A).

Cytological samples were obtained from 25 dogs (Appendix A); 17 patients provided 1 case each, 6 patients provided 2 cases each, 1 patient provided 3 cases, and 1 patient provided 8 cases.

All details concerning dogs signalment, LN localization, and pathological process (NOD and MCT) are provided in Appendix A.

### 3.2. Nodal Mast Cells in Non-Oncological Dogs (NODs) and Mast Cell Tumor-Bearing Dogs (MCTBDs)

The median range of nodal MC absolute number was 0–0.5 and 3.90–28 for NODs and MCTBDs, respectively; the mean range was 0–1.7 and 79.22–428.3 for NODs and MCTBDs, respectively (details for each counting method are reported in Appendix A). The error coefficient for each counting method applied to MCTBDs ranged from 0.28 to 0.31 (Appendix A).

MCTBDs had a significantly higher number of MCs than NODs (*p* < 0.0001), regardless of the counting method applied (Figure 1).

Although not statistically significant, mean cellularity was higher in cases obtained from NODs (median: 286.7) than from MCTBDs (median: 222.2). Spearman correlation analysis revealed a significant inverse correlation between MC number and cellularity in samples obtained from MCTBDs (r range: −0.4112 to −0.5281; p value range: 0.0027 to 0.0240), but not for samples obtained from NODs (r range: 0.03906 to 0.05803; *p* value range: 0.9212 to >0.9999).

### 3.3. Nodal Mast Cells in Specimens Obtained from Mast Cell Tumor-Bearing Dogs (MCTBDs) and Classified as “Non Metastatic” (MCT-NM), “Possibly Metastatic” (MCT-PM), and “Metastatic” (MCT-M)

Based on the metastatic status, as determined according to criteria reported in Table 1, 10 specimens from MCTBDs (10/30) were classified as MCT-NM, 5 as MCT-PM, and 15 as MCT-M.

Histopathological evaluation of the nodal metastatic status according to Weishaar et al. [17] was available in 23 cases (23/30) (Appendix A). Cytological and histological diagnosis completely or partially agreed in 15/23 cases (65.22%) and in 4/23 cases (17.39%), respectively (Appendix A). No agreement was observed in 4/23 cases (17.39%).

When considering the different counting methods applied, the median range of nodal MC absolute number was 0–2 for MTC-NM cases, 5–28 for MTC-PM cases, and 120–782 for MTC-M cases, respectively. The mean range of nodal MC absolute number was 0.25–2.40, 4.05–19.40, and 142.7–718.3 for MTC-NM, MTC-PM, and MTC-M cases, respectively (Appendix A).

When the number of MCs was compared with nodal metastatic status, applying the Kruskall-Wallis test with Dunn’s correction for multiple comparisons, cases classified as MCT-M constantly showed a significantly higher number of MCs (*p* < 0.0001) than MCT-NM cases, but not compared to MCT-PM cases (Figure 2), regardless of the counting method applied. Additionally, no significant difference in the number of nodal MCs was observed between MCT-NM and MCT-PM cases.

Despite not being significantly different, mean cellularity showed a decreasing trend, being slightly higher in MCT-NM cases (median: 251.8) compared to MCT-PM ones (median: 238.6), which in turn were characterized by increased cellularity compared to MCT-M cases (median: 162.2; Appendix A). Spearman analysis revealed a significant inverse correlation between MC number and cellularity in cases classified as MCT-M (r range: −0.5536 to −0.7500; *p* value range: 0.0014 to 0.0349), which instead was not observed in MCT-NM (r range: −0.3206 to 0.6272; *p* value range: 0.0591 to >0.9999) and MCT-PM cases (r range: −0.3000 to 0.4617; *p* value range: 0.433 to 0.9500).

### 3.4. Nodal Mast Cells in Specimens Obtained from Mast Cell Tumor-Bearing Dogs (MCTBDs) and Sampled via Fine-Needle Aspiration (MCT-F), Scraping Smearing (MCT-S), and Touch Imprinting (MCT-T)

Among MCT-F, 4/10 were classified as MCT-NM, 1 as MCT-PM, and 5 as MCT-M. Among MCT-S, 2/10 were classified as MCT-NM, 2 as MCT-PM, and 6 as MCT-M. Among MCT-T, 4/10 were classified as MCT-NM, 2 as MCT-PM, and 4 as MCT-M. (Appendix A).

The median range of nodal MC absolute number was 94.5–620 for MCT-F, 19.10–136.5 for MCT-S, and 2.25–17.5 for MCT-T. The mean range of nodal MC absolute number was 156.3–833.6 for MCT-F, 57.83–365.8 for MCT-S, and 14.66–85.6 for MCT-T. Details regarding the number of nodal MCs for each counting method are reported in Appendix A.

The Kruskall-Wallis test with Dunn’s correction for multiple comparisons did not reveal any statistically significant difference among the groups when different sample preparation techniques were compared, irrespective of the counting method applied. Nonetheless, a descending trend indicative of higher numbers of MCs in FNAs was observed, followed in order, by cases obtained by scraping smearing and touch imprinting (Figure 3).

Mean cellularity showed an increasing trend, being lower in MCT-F (median: 147.2) compared to MCT-S (median: 245.8) and slightly lower in MCT-S compared to MCT-T (median: 281.7; Appendix A). The Kruskall-Wallis test with Dunn’s correction for multiple comparisons revealed a significant difference in the mean cellularity between MCT-F and MCT-T (*p* < 0.01).

Spearman correlation analysis revealed a significant inverse correlation between MC number and cellularity in MCT-S with all counting methods (r range: −0.6525 to −0.8182; p value range: 0.0047 to 0.0463) except for counting of MCs in 8 HPFs (r: −0.6364; *p* value: 0.0544), and between MC number and cellularity in MCT-F exclusively for counting over 500 cells (r: −0.6930; *p* value: 0.0314), although the Spearman coefficient was also constantly negative for the other counting methods (r range: −0.4085 to −0.6930; *p* value range: 0.0679 to 0.2153). On the contrary, Spearman correlation analysis did not reveal any significant correlation between MC number and cellularity in MCT-T, regardless of the counting method applied for MC quantification (r range: −0.006079 to 0.1376; *p* value range: 0.7026 to 0.9924).

## 4. Discussion

The current study aimed to verify whether the cytological quantification of nodal MCs is useful per se in determining the nodal metastatic status in dogs with MCT, and to verify whether the cytological sample preparation technique influences the number of MCs in nodal cytological specimens obtained from MCTBDs.

Cytological LN specimens obtained from MCTBDs were characterized by a significantly higher number of MCs compared to NODs, according to the previous literature [9,14,15,16,18,20,21,23,27,48].

However, differences in MCs number were significant only between MCT-NM and MCT-M, suggesting that MCs quantification on cytological samples is not sufficient per se to distinguish among non-metastatic, possibly metastatic, and metastatic cytological samples. This finding is only partially in agreement with previous studies that reported an increasing trend in the number of nodal MCs shifting from non-metastatic cases to metastatic ones [9,23]. The lack of significant differences between MCT-NM and MCT-PM cases in the present study might be a consequence of the low number of cases classified as MCT-PM as well as of the cytological classification criteria applied [15,21]. Indeed, the criteria for inclusion in the class MCT-PM are based on the presence of two or three pairs or triplets of MCs on an entire slide, rather than on MC number; therefore, it can be hypothesized that the uneven distribution of aggregated MCs among microphotographs might have influenced our results. Additionally, comparison with previous studies might have been further limited by the inclusion of different sample preparation techniques in the current study.

Former studies that focused on the cytological quantification of nodal MCs in healthy dogs, dogs affected by non-neoplastic diseases (e.g., allergies), and dogs with MCT [9,21,23,30,32,33] applied different inclusion criteria, counting methods, and statistical analysis, limiting the comparison of their findings with our results.

To the best of our knowledge, only four studies included the quantification of MCs in cytological LN samples from MCT-bearing dogs [9,21,23,30].

Specifically, Mutz et al. [21] compared MC estimates over 2000 cells in LNs obtained from healthy dogs, dogs with allergic dermatopathy, and MCT-bearing dogs. While in Mutz’s study, results regarding MCT-bearing dogs almost overlapped with ours, MC estimates in allergic dogs were higher than those in NODs in the current study. Additionally, Mutz and coauthors observed that MCT-bearing dogs were more likely to present at least one nodal MC or >0.05% nodal MCs compared to healthy, but not when compared with allergic dogs. Conversely, in the present study, MCTBDs showed a significantly higher number of nodal MCs compared to NODs. These differences might be the consequence of the heterogeneity of pathological conditions affecting NODs included in our study.

Marconato et al. [9] evaluated LN cytological specimens obtained from healthy dogs, dogs with infectious or inflammatory diseases, and MCT-bearing dogs, with the last further subdivided into dogs without LN metastases, dogs with LN “inconclusive” for metastasis, and dogs with LN metastasis. Nodal MC estimates observed in the current study were consistent with those reported by these authors for NODs, MCTBDs without LN metastasis, and MCTBDs with LN metastasis. On the other hand, we observed a lower number of nodal MCs in MCT-PM LNs compared to MCTBDs with LN inconclusive for metastasis evaluated by Marconato et al. This difference might be the consequence of the different cytological evaluation criteria applied. Indeed, in the current study, the cytological interpretative system was mainly based on the evaluation of MC tendency to aggregate rather than on cells quantification as done by Marconato et al. [9]. Those authors also reported that MCT-bearing dogs with LN metastasis had a significantly higher number of nodal MCs compared to healthy dogs, dogs affected by non-neoplastic infectious-inflammatory conditions, and MCT-bearing dogs without LN metastasis and with LN inconclusive for metastasis, in partial agreement with the present results.

Sulce et al. [23] applied the cytological interpretative system proposed by Krick et al. [15] to investigate the number of nodal MCs over 1000 cells in 12 LNs obtained from MCTBDs. While MC estimates reported for LN samples classified as “reactive lymphoid hyperplasia” and “probable” or “certain metastasis” overlapped almost perfectly with those reported in the current study, our results for MCT-PM LNs were markedly lower than those observed by Sulce et al. This latter finding might be the consequence of the different number of cases evaluated in each study, or of the difference in sample preparation techniques applied. Alternatively, our lower number of MCs might be due to the fact that 4/5 LNs classified as MCT-PM in the current study were clinically non-palpable/normal-sized SLN (Appendix A), suggesting that the metastatic process might have been in its first phases. However, no data regarding the size of LNs investigated were reported by Sulce et al. and, therefore, this hypothesis cannot be verified.

Finally, Sabattini et al. [30] applied semi-quantitative estimation of MCs in LN cytological samples obtained from MCT-bearing dogs and histologically classified as HN2 or HN3. MC estimates were evaluated on the entire slides and only aggregated data were reported in the paper, thus making the comparison with our results not possible.

To the best of our knowledge, this is the first study quantifying MCs in cytological nodal specimens from MCTBDs sampled with different techniques. The differences in the number of MCs among the different sample preparation techniques were not statistically significant, suggesting that the sample preparation technique does not influence the cytological evaluation of LN metastatic status in MCT-bearing dogs. This result opens the road to future investigations on the intraoperative use of cytology for the real-time evaluation of surgically extirpated SLNs. Future studies on a larger cohort of cases, possibly applying all three different methods to the same lymph node, are advisable to confirm our data.

Additionally, it is noteworthy that the cytological diagnoses obtained from the same lymph node by scraping smear and by touch imprint were not consistent in three cases (Appendix A). Specifically, in two cases, the histopathological diagnosis was underestimated in scraping smears, while in the other case, a partially correct diagnosis was provided for the touch imprint. This finding underlies the need for further studies aiming to validate the diagnostic accuracy of cytology in general in this field. It should also be considered that histopathology is currently the gold standard for the evaluation of lymph node metastatic status and, therefore, it is used as a comparison to establish the diagnostic accuracy of cytology. However, histopathology providing a two-dimensional assessment of a three-dimensional structure may not be fully representative of the nodal metastatic status; therefore, a further diagnostic accuracy study should include methods to mitigate this bias.

A decreasing trend in the number of MCs from FNAs to touch imprints has been observed and might be explained by potential correlations between the size of sampled LNs and their metastatic status. Specifically, at least 8/10 of MCT-F were reported as megalic (Appendix A), among which one was classified as MCT-PM and three as MCT-M. It is, therefore, likely that in these cases, lymph-adenomegaly was determined by the invasion of the LNs by metastasis, with a consequent marked increase in the number of MCs. On the contrary, most of the MCT-S or MCT-T cases were obtained from clinically non-palpable/normal-sized SLNs. This may have led to a surgical extirpation for those LNs classified as “metastatic” at an earlier stage of metastatic invasion, prior to lymph-adenomegaly development and with less MCs infiltration.

When the mean cellularity was investigated between the groups evaluated in the current study, the only significant differences were observed between MCT-F and MCT-T. The lower cellularity of scraping smears compared to touch imprints could be related to the greater number of broken cells resulting from the traumatic action on the neoplastic cells during smear preparation, although the literature reports that the scraping smears should provide an increased cellularity compared to FNA [49,50]. Additionally, a significant inverse correlation between estimates of nodal MCs and mean cellularity was almost constantly observed for MCTBDs in general, for MCT-M cases and for MCT-S, disregarding the counting method applied. These significant correlations observed might be the result of the number of MCs in the lymph node. Indeed, it is conceivable that MCs, being larger than lymphocytes, might reduce the space for other cells leading to a lower mean cellularity.

The statistically significant differences observed among the groups were maintained irrespectively of the counting method applied for MC quantification. Counting methods “over cells” are still considered more reliable because they are based on a proportionality ratio that is not theoretically affected by the extremely variable cellularity that can characterize cytological samples [9,21]. Nonetheless, our findings open the road to the application of counting methods “per field” in further investigations, as supported by an almost constant error coefficient among the different counting methods when applied to the MCTBDs group in its entirety. Counting “per fields” might indeed markedly reduce the time needed for MC quantification compared to counting methods “over cells”. Other strengths of the current study are the application of numerous different counting methods, which could improve the comparison with potential future studies, as well as the quantification of MCs in microphotographs rather than directly at the microscope, which could improve the standardization of future interrater agreement investigations.

The lack of systematical comparison of the cytological evaluation of the LN metastatic status with the corresponding histological diagnosis and of repeated evaluation by multiple investigators represent the major limitations of the current work.

The retrospective nature of the study and the experimental design determined a few further limitations. For instance, the different number of cases assigned to each of the subcategories according to the cytological diagnosis may have influenced the statistical analysis.

## 5. Conclusions

In conclusion, although MCs counting might be an objective method to evaluate nodal metastasis in MCTBDs, this technique seems not to be useful per se to distinguish among non-metastatic, possibly metastatic, and metastatic cytological samples. Nonetheless, according to the observation that the system by Krick et al. [15] is mainly based on morphological and aggregation criteria rather than on the quantification of MCs, further investigations aiming to evaluate the influence of MCs quantification on the final diagnostic accuracy of cytology compared to the reference test (i.e., histological evaluation) are warranted. Although limited by the lack of application of all three different sample preparation techniques on each LN, our findings suggest that sample preparation technique does not influence the number of nodal MCs. Therefore, future studies aiming to test the diagnostic accuracy of cytology could include specimens obtained with different sample preparation technique, theoretically without biasing the results. Finally, it is noteworthy that all the differences found in this study were always maintained between the different groups investigated regardless of the counting method applied. This observation opens the road to the use of counting methods “per fields” in future studies, which could significantly speed up cytological evaluation times compared to counting methods “over cells”.

## Figures and Tables

**Figure 1 animals-13-02634-f001:**
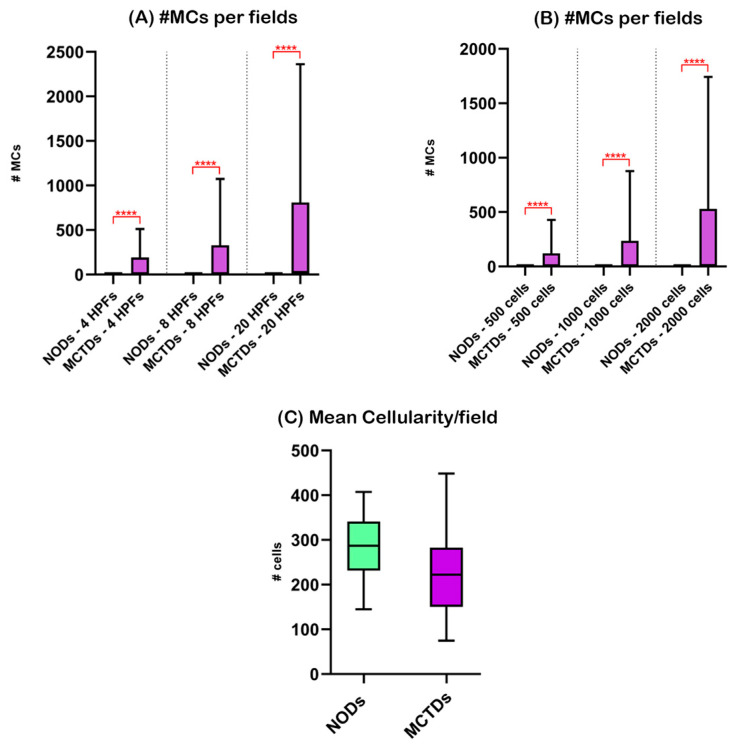
Nodal mast cells and sample cellularity in non-oncological dogs (NODs) and mast cell tumor-bearing dogs (MCTBDs). The box represents the interquartile range, with the internal horizontal line indicating the median value. The tip of the upper whisker indicates the maximum value, while the tip of the lower whisker reports the minimum value. Statistically significant differences as determined with Mann-Whitney test are indicated by red bars and stars. HPFs, high power fields; MCs, mast cells; # number of; **** *p* < 0.0001.

**Figure 2 animals-13-02634-f002:**
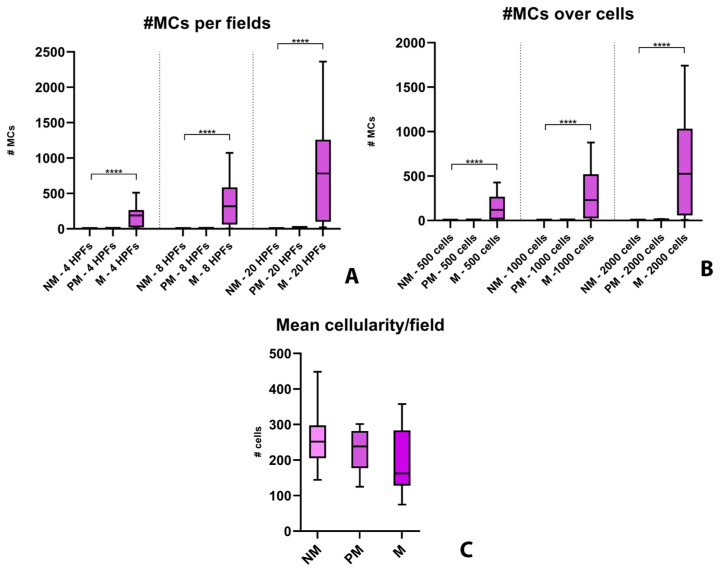
Nodal mast cells and sample cellularity referring to specimens obtained from mast cell tumor-bearing dogs (MCTBDs) and classified as “non metastatic” (NM), “possibly metastatic” (PM), and “metastatic” (M). The box represents the interquartile range, with the internal horizontal line indicating the median value. The tip of the upper whisker indicates the maximum value, while the tip of the lower whisker reports the minimum value. Statistically significant differences as determined with Kruskall-Wallis test with Dunn’s correction for multiple comparisons are indicated by black bars and stars. Legend: HPFs, high power fields; MCs, mast cells; # number of; **** *p* < 0.0001.

**Figure 3 animals-13-02634-f003:**
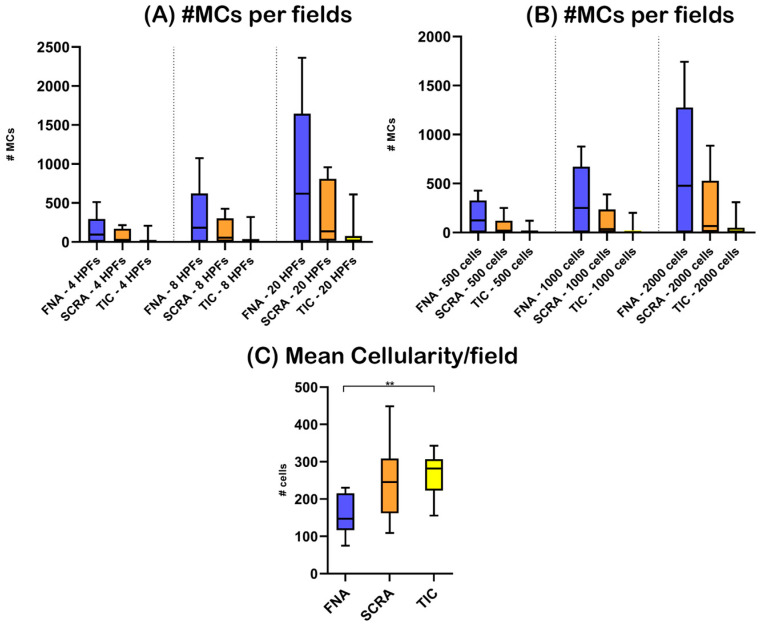
Nodal mast cells and sample cellularity referring to specimens obtained from mast cell tumor-bearing dogs (MCTBDs) and sampled via fine-needle aspiration (FNA), scraping smearing (SCRA), and touch imprinting (TIC). The box represents the interquartile range, with the internal horizontal line indicating the median value. The tip of the upper whisker indicates the maximum value, while the tip of the lower whisker reports the minimum value. Statistically significant differences as determined with Kruskall-Wallis test with Dunn’s correction for multiple comparisons, are indicated by black bars and stars. Legend: HPFs, high power fields; MCs, mast cells; #, number; ** *p* < 0.01.

**Table 1 animals-13-02634-t001:** Criteria and corresponding interpretation applied for the cytological evaluation of LN metastatic status in MCT-bearing dogs.

Cytological Criteria	Interpretation
No MCs observed; OR >50% small lymphocytes with a mixed population of prolymphocytes, lymphoblasts, plasma cells, and/or few to moderate numbers of macrophages, neutrophils, and eosinophils, and/or rare individual MCs	**Non-metastatic (MCT-NM)** *(former “normal” + “reactive lymphoid hyperplasia”)*
2–3 incidences of MCs aggregated in couples or triplets	**Possibly metastatic (MCT-PM)**
>3 incidences of MCs aggregated in couples or triplets and/or 2–5 aggregates composed by >3 MCs; OR effacement of lymphoid tissue by MCs, and/or aggregated, poorly differentiated MCs (pleomorphism, anisocytosis, anisokaryosis, and/or decreased or variable granulation), and/or >5 aggregates composed by >3 MCs	**Metastatic (MCT-M)** *(former “probable metastasis” + “certain metastasis”)*

Table adapted from Krick et al. [15] as modified by Mutz et al., 2017; and Fournier et al., 2020 [21,36]. MC, mast cell.

## Data Availability

All the raw data presented in this study are available in Appendix A.

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
