# Peer review of "Cytological Quantification of Nodal Mast Cells in Dogs Affected by Non-Neoplastic Condition and Mast Cell Tumor Using Different Sample Preparation Techniques: An Explorative Study"

_animals, 2023, doi:10.3390/ani13162634_

Round 1

Reviewer 1 Report

Reviewer comments to the manuscript ” CYTOLOGICAL QUANTIFICATION OF NODAL MAST CELLS IN DOGS AFFECTED BY NON NEOPLASTIC CONDITION AND MAST CELL TUMOR USING DIFFERENT SAMPLING TECHNIQUES: AN EXPLORATIVE STUDY” by Buzzi et at., submitted to “Animals” (2023)

In their manuscript, the authors analyzed the potential and limits of cytological evaluation of lymph nodes during staging of canine mast cell tumors using different modes of sample preparation and (semi)quantitative analysis of mast cell numbers.

The study is all over well designed, the pertinent literature is adequately summarized in the introduction, materials and methods are orderly described, results are well represented in text and graphs, and the discussion basically addresses all relevant aspects of the study.

 Although the findings of the study – i.e., that the numbers of counted mast cells in samples of dogs with mast cell tumors with and without metastasis are higher than in non-tumor bearing animals and that the applied mode of sample preparation (FNA, scrapings, imprints) and (semi)quantitative analysis of mast cell numbers (counting of cells in FOV (= defined areas) or as quotients of total counted cell numbers) do not differentially affect the results of the “counting” process are not astonishing, these results are –in my opinion- worth publishing and I consider them of general interest to the readership of “Animals”.

I would suggest to introduce a few changes to the manuscript, which could further contribute to increase its clearity:

1.) Please avoid usage of the term “sampling technique” without a clear statement that this term as it is used in the manuscript text only refers to the technical procedure of transferring tissue/cells to the cytology slide (i.e. FNA, scraping, imptints), but not to a sampling procedure in terms of a selection/decision process to choose the sampled locations from a defined reference compartment (here: lymph node), such as  e.g., random sampling procedures, systematic uniform random sampling procedures, or sampling from anatomically defined sampling sites. It must made clear, that here, arbitrarily chosen locations of lymph nodes have been sampled and analyzed.

2.) Due to the applied mode of sample preparation, the representativity of the examined cytological slides for the reference compartment (i.e., the overall cellular composition within the entire lymph node) cannot be assumed- this is of course a general situation in cytology samples, nevertheless it should be explicitly mentioned. In the discussion section, the authors absolutely correctly state this (line 389): “This finding underlies the need for further studies aiming to validate the diagnostic accuracy of cytology in general in this field”. However, a relation to histology alone (i.e., a 2D section from a 3D lymph node) will not be sufficient for this – a thorough analysis would require quantitative morphological analyses of cell numbers (numerical volume densities) in the entire reference compartment (here: lymph node) with appropriate representative sampling methods by unbiased quantitative stereological analyses. Please mention this.

3.) Irrespective of a generally questionable representativity of the analyzed samples for the reference compartment, the representativity of the analyzed slide-areas for the entire samples, as well as the reproducibility of the “counting” results in repeated cell-counts in the same, as well as in different slide areas of the same slide should be addressed in more detail. The applied mode of sampling of FOV in the and the effects of different sampling approaches slides (SUR-sampling, random sampling, arbitrarily selected FOVS, …) on the reproducibility of the results and the effectiveness of the analyses should be discussed. The presented data generally show a high variability of counting values in the different groups. The authors should therefore consider to make some efforts to provide information on the contribution of the inter-individual variability, intra-sample variability (depending on the sampling regime used to determine the analyzed FOVs in the slides), and the technically introduced variability to the overall observed variance of the results by appropriate statistical parameters, such as error coefficients.

4.) In line 410, the authors hypothesize that “….hat MCs, being larger than lymphocytes, might reduce the space for other cells leading to a lower mean cellularity” – This is not a hypothesis but a simple consequence of geometry –i.e., that equal numbers of large and small objects will have different volumes -and in a monolayer of cells, larger cells will naturally occupy a  larger area (the same is of course also true for histology sections: larger objects/cells do generally have a higher random probability to get sectioned -just because of being larger- than small objects/cells and therefore have a higher chance to be represented in the section – and will therefore be overrepresented in a section as compared to smaller objects with the same numerical volume density…) – Therefore, “counting per field” instead of “counting over cells” cannot be used to relate proportions of different objects/cell types with different volumes (!) - not in histolugy and also not in cytology slides- and of course always needs to consider effects of cell number overestimation due to edge effects (e.g., by using counting frames with in- and exclusion lines). The authors should consider this, when favoring “counting per” fields instead of “counting over cells” (see line. 417 et seq., line 446 et seq.). Strictly, also the sampling fraction (i.e., the proportion of the examined sample relative to the total volume of the sampled reference compartment) should be equal in all examined specimen of a study.

5.) In my opinion, the study might also benefit from some representative cytology images, e.g., showing overviews of the FOVs within cytology slides.

Author Response

Reviewer comments to the manuscript ” CYTOLOGICAL QUANTIFICATION OF NODAL MAST CELLS IN DOGS AFFECTED BY NON NEOPLASTIC CONDITION AND MAST CELL TUMOR USING DIFFERENT SAMPLING TECHNIQUES: AN EXPLORATIVE STUDY” by Buzzi et at., submitted to “Animals” (2023).

In their manuscript, the authors analyzed the potential and limits of cytological evaluation of lymph nodes during staging of canine mast cell tumors using different modes of sample preparation and (semi)quantitative analysis of mast cell numbers.

The study is all over well designed, the pertinent literature is adequately summarized in the introduction, materials and methods are orderly described, results are well represented in text and graphs, and the discussion basically addresses all relevant aspects of the study.

 Although the findings of the study – i.e., that the numbers of counted mast cells in samples of dogs with mast cell tumors with and without metastasis are higher than in non-tumor bearing animals and that the applied mode of sample preparation (FNA, scrapings, imprints) and (semi)quantitative analysis of mast cell numbers (counting of cells in FOV (= defined areas) or as quotients of total counted cell numbers) do not differentially affect the results of the “counting” process are not astonishing, these results are –in my opinion- worth publishing and I consider them of general interest to the readership of “Animals”.

I would suggest to introduce a few changes to the manuscript, which could further contribute to increase its clearity:

1.) Please avoid usage of the term “sampling technique” without a clear statement that this term as it is used in the manuscript text only refers to the technical procedure of transferring tissue/cells to the cytology slide (i.e. FNA, scraping, imptints), but not to a sampling procedure in terms of a selection/decision process to choose the sampled locations from a defined reference compartment (here: lymph node), such as  e.g., random sampling procedures, systematic uniform random sampling procedures, or sampling from anatomically defined sampling sites. It must made clear, that here, arbitrarily chosen locations of lymph nodes have been sampled and analyzed.

Thank you for your comment. The term “sampling technique” has been substituted with “sample preparation technique” throughout the manuscript and in the title.

2.) Due to the applied mode of sample preparation, the representativity of the examined cytological slides for the reference compartment (i.e., the overall cellular composition within the entire lymph node) cannot be assumed- this is of course a general situation in cytology samples, nevertheless it should be explicitly mentioned. In the discussion section, the authors absolutely correctly state this (line 389): “This finding underlies the need for further studies aiming to validate the diagnostic accuracy of cytology in general in this field”. However, a relation to histology alone (i.e., a 2D section from a 3D lymph node) will not be sufficient for this – a thorough analysis would require quantitative morphological analyses of cell numbers (numerical volume densities) in the entire reference compartment (here: lymph node) with appropriate representative sampling methods by unbiased quantitative stereological analyses. Please mention this.

Thank you for your remark. We absolutely agree that our models, and specifically those models that are applied and appliable in routine diagnostic, are often not representative of the reality. We added a few lines to the text (lines 414-420) to mention this aspect.

3.) Irrespective of a generally questionable representativity of the analyzed samples for the reference compartment, the representativity of the analyzed slide-areas for the entire samples, as well as the reproducibility of the “counting” results in repeated cell-counts in the same, as well as in different slide areas of the same slide should be addressed in more detail. The applied mode of sampling of FOV in the and the effects of different sampling approaches slides (SUR-sampling, random sampling, arbitrarily selected FOVS, …) on the reproducibility of the results and the effectiveness of the analyses should be discussed. The presented data generally show a high variability of counting values in the different groups. The authors should therefore consider to make some efforts to provide information on the contribution of the inter-individual variability, intra-sample variability (depending on the sampling regime used to determine the analyzed FOVs in the slides), and the technically introduced variability to the overall observed variance of the results by appropriate statistical parameters, such as error coefficients.

Thank you for your observation, we provided additional data concerning error coefficients (material and methods lines 173-175; results lines 206-207 and discussion lines 440-441). Please see also supplementary table 2. Moreover a comment concerning repeated cell-counts and results reproducibility has been added (lines 448-450)

4.) In line 410, the authors hypothesize that “….hat MCs, being larger than lymphocytes, might reduce the space for other cells leading to a lower mean cellularity” – This is not a hypothesis but a simple consequence of geometry –i.e., that equal numbers of large and small objects will have different volumes -and in a monolayer of cells, larger cells will naturally occupy a  larger area (the same is of course also true for histology sections: larger objects/cells do generally have a higher random probability to get sectioned -just because of being larger- than small objects/cells and therefore have a higher chance to be represented in the section – and will therefore be overrepresented in a section as compared to smaller objects with the same numerical volume density…) – Therefore, “counting per field” instead of “counting over cells” cannot be used to relate proportions of different objects/cell types with different volumes (!) - not in histolugy and also not in cytology slides- and of course always needs to consider effects of cell number overestimation due to edge effects (e.g., by using counting frames with in- and exclusion lines). The authors should consider this, when favoring “counting per” fields instead of “counting over cells” (see line. 417 et seq., line 446 et seq.). Strictly, also the sampling fraction (i.e., the proportion of the examined sample relative to the total volume of the sampled reference compartment) should be equal in all examined specimen of a study.

Thank you for you observation. We agree with you that this is simple geometry, and not a new hypothesis. We incorrectly used this term since we could not find any references that support our observation. However, this could be a misleading term, suggesting that we actually formulate a new hypothesis, therefore we slightly modified the sentence amending the word “hypothesis”.

5.) In my opinion, the study might also benefit from some representative cytology images, e.g., showing overviews of the FOVs within cytology slides.

Thank you for your remark, a figure has been added (supplementary figure 2 – line 157) showing FOVs and examples of selection and exclusion of cells.

Reviewer 2 Report

Comments on the manuscript animals-2525812-peer-review-v1 entitled “Cytological quantification of nodal mast cells in dogs affected by non neoplastic condition and mast cell tumor using different sampling techniques: an explorative study” by Buzzi et al.

This is an interesting article on the importance of cytological evaluation on mast cell tumour staging. As mast cells are a common cell in non-neoplastic lymph nodes, it becomes challenging to determine if these cells represent micrometastatic deposits, or a non-pathologic condition. If it is easy to detect macrometastasis, identify micrometastasis by cytology is very difficult, and if results are affected by different sample collection techniques and/or different microscopic fields, it is important to stablish. Furthermore, as mentioned by authors, standardization on procedures is imperative either on clinical practice and investigation. The major criticism I have is the fact that authors use both the mean value and median value of cell counts, turning results difficult to analyze and understand. However, results are interesting, and articles on this theme are welcome. There are some minor changes that should be amended.

Though results are as expected, in the MC counts in NODs and MCTDs, in my opinion it is surprising that the mean counts of NODs were higher than in MCTDs. Authors justified this, but I never saw, in the literature a similar report. It´s plausible, but… 

It is also surprising the low cell counts of impression smears or scraping: as authors justify, probably in these techniques there are a bigger cell destruction. But it is surprising…In my opinion, these points should be more explored in the discussion section, and authors should proposed the technique they think that allows a more accurate results for the practical clinician.

SIMPLE SUMMARY

Line 14: In the sentence, replace “…of canine MCT…” to “…of canine Mast Cell Tumor (MCT)…”

Line 16: Put “in vitro” in italic.

Lines 19-20: Please replace “…Our results suggested that…” to “…Our results suggests that …”

ABSTRACT:

Line 29: There´s an extra space in the sentence “…in : non-metastatic…”. Please amend.

Line 30-31: Please clarify the sentence “MCs were counted in 4, 8, and 20 high-power-fields, and over 500, 1000, and 2000 cells.”- authors counted over 500, 1000, and 2000 of total cells not MC. 

Line 32-33: Please replace “…There was no significant difference “metastatic” and “possibly metastatic” samples …”. To “…There was no significant difference between “metastatic” and “possibly metastatic” samples …”.

INTRODUCTION SECTION

There are multiple dots before the references (as in lines 51-52, 57, 63, and so on). Please amend.

MATERIALS AND METHODS SECTION

Line 121: In the sentence …” (i.e. MCT-F MCT-S and MCT-T)”, there´s a comma missing. Please amend.

Lines 245-246: In the sentence “…during daily clinical activities.” Is in italic. Please amend.

Table 1: Plasma Cells is not correctly written. Please amend.

In the legend of table (lines 137-138), remove the word “Legend”. Replace “MC, mast cell. Table adapted from Krick et al.[15] as modified by Mutz et 137 al., 2017; and Fournier et al., 2020 [21,36].” To “Table adapted from Krick et al.[15] as modified by Mutz et 137 al., 2017; and Fournier et al., 2020 [21,36]. MC, mast cell.

Why authors distinguish “Possibly metastatic (MCT-PM)” from “former “possible metastasis”- is not the same??

RESULTS SECTION

Line 188: Please amend the sentence ”…cases. (Supplementary Figure 1)” to “cases (Supplementary Figure 1).

Line 199: Please amend the sentence “… respectively. (Details for each counting method are reported in Supplementary Table 2).” to “… respectively (Details for each counting method are reported in Supplementary Table 2).

Figures

The figure legend should be always after the figures. Remove the word “legend” before “HPFs” in all figures legends.

As example:…” FIGURE 1. Nodal mast cells and sample cellularity in non-oncological dogs (NODs) and mast cell tumor-bearing dogs (MCTDs). 

The box represents the interquartile range, with the internal horizontal line indicating the 

median value. The tip of the upper whisker indicates the maximum value while the tip of the lower 216 whisker reports the minimum value. Statistically significant differences as determined with 217 Mann-Withney test are indicated by red bars and stars. HPFs, high power fields; MCs, 218 mast cells; #, number of; ****, p<0.0001.”

In figure 2, identify the figures as A, B, and C.

DISCUSSION SECTION

There are multiple dots before the references (as in lines 316-317, 323, and so on). Please amend.

REFERENCES

There are some journal names in full and others in short (as ref 4, 13, 23, 35, and 41). Please uniform the criteria.

Author Response

This is an interesting article on the importance of cytological evaluation on mast cell tumour staging. As mast cells are a common cell in non-neoplastic lymph nodes, it becomes challenging to determine if these cells represent micrometastatic deposits, or a non-pathologic condition. If it is easy to detect macrometastasis, identify micrometastasis by cytology is very difficult, and if results are affected by different sample collection techniques and/or different microscopic fields, it is important to stablish. Furthermore, as mentioned by authors, standardization on procedures is imperative either on clinical practice and investigation. The major criticism I have is the fact that authors use both the mean value and median value of cell counts, turning results difficult to analyze and understand. However, results are interesting, and articles on this theme are welcome. There are some minor changes that should be amended.

Though results are as expected, in the MC counts in NODs and MCTDs, in my opinion it is surprising that the mean counts of NODs were higher than in MCTDs. Authors justified this, but I never saw, in the literature a similar report. It´s plausible, but… 

It is also surprising the low cell counts of impression smears or scraping: as authors justify, probably in these techniques there are a bigger cell destruction. But it is surprising…In my opinion, these points should be more explored in the discussion section, and authors should proposed the technique they think that allows a more accurate results for the practical clinician.

Thank you for your observations. We agree that some results were somehow unexpected, however they can also be explained as you outlined. Since this study is explorative, we believe it is perhaps premature to indicate a technique of choice for lymph-nodes examination in MCT affected dogs. Moreover, impression smear and scraping can not be easily applied in routine in vivo examination of lymph-nodes and, have been used here mainly for comparison in view of application of cytology for intraoperative evaluation of surgically excised lymph nodes.

SIMPLE SUMMARY

Line 14: In the sentence, replace “…of canine MCT…” to “…of canine Mast Cell Tumor (MCT)…”

Line 16: Put “in vitro” in italic.

Lines 19-20: Please replace “…Our results suggested that…” to “…Our results suggests that …”

 Thank you for your observations, all the proposed corrections have been done

ABSTRACT:

Line 29: There´s an extra space in the sentence “…in : non-metastatic…”. Please amend.

Line 30-31: Please clarify the sentence “MCs were counted in 4, 8, and 20 high-power-fields, and over 500, 1000, and 2000 cells.”- authors counted over 500, 1000, and 2000 of total cells not MC. 

Line 32-33: Please replace “…There was no significant difference “metastatic” and “possibly metastatic” samples …”. To “…There was no significant difference between “metastatic” and “possibly metastatic” samples …”.

Thank you for your observations, all the proposed corrections have been done

INTRODUCTION SECTION

There are multiple dots before the references (as in lines 51-52, 57, 63, and so on). Please amend.

We apologize for the typing error; the multiple dots have been amended 

MATERIALS AND METHODS SECTION

Line 121: In the sentence …” (i.e. MCT-F MCT-S and MCT-T)”, there´s a comma missing. Please amend.

Lines 245-246: In the sentence “…during daily clinical activities.” Is in italic. Please amend.

Thank you for your observations, all the proposed corrections have been done

Table 1: Plasma Cells is not correctly written. Please amend.

In the legend of table (lines 137-138), remove the word “Legend”. Replace “MC, mast cell. Table adapted from Krick et al.[15] as modified by Mutz et 137 al., 2017; and Fournier et al., 2020 [21,36].” To “Table adapted from Krick et al.[15] as modified by Mutz et 137 al., 2017; and Fournier et al., 2020 [21,36]. MC, mast cell.

Why authors distinguish “Possibly metastatic (MCT-PM)” from “former “possible metastasis”- is not the same??

Thank you for your observations, all the proposed corrections have been done

As far as the last consideration is concerned, “possibly metastatic” is the term used in the modified criteria. It is mostly a matter of words. But we agree that it can be misleading, we therefore amended the sentence “former possible metastatic”.

RESULTS SECTION

Line 188: Please amend the sentence ”…cases. (Supplementary Figure 1)” to “cases (Supplementary Figure 1).

Line 199: Please amend the sentence “… respectively. (Details for each counting method are reported in Supplementary Table 2).” to “… respectively (Details for each counting method are reported in Supplementary Table 2).

Thank you for your observations, all the proposed corrections have been done

Figures

The figure legend should be always after the figures. Remove the word “legend” before “HPFs” in all figures legends.

As example:…” FIGURE 1. Nodal mast cells and sample cellularity in non-oncological dogs (NODs) and mast cell tumor-bearing dogs (MCTDs). 

The box represents the interquartile range, with the internal horizontal line indicating the 

median value. The tip of the upper whisker indicates the maximum value while the tip of the lower 216 whisker reports the minimum value. Statistically significant differences as determined with 217 Mann-Withney test are indicated by red bars and stars. HPFs, high power fields; MCs, 218 mast cells; #, number of; ****, p<0.0001.”

In figure 2, identify the figures as A, B, and C.

Thank you for your observations, all the proposed corrections have been done. A, B; C have been indicated in figure 2.

DISCUSSION SECTION

There are multiple dots before the references (as in lines 316-317, 323, and so on). Please amend.

We apologize for the typing error; the multiple dots have been amended 

REFERENCES

There are some journal names in full and others in short (as ref 4, 13, 23, 35, and 41). Please uniform the criteria.

We apologize for the mistake, in the present version of the manuscript journal names have been uniformed (all in full)